# Analyzing the Inlet Gas Void Fraction on the Flow Characteristics for a Multiphase Pump Based on Cavitation Cases

**Wenjuan Lv [1,2,*], Haigang Wen [1,2], Guangtai Shi [1,2] and Shan Wang [1]**

[1]  Key Laboratory of Fluid and Power Machinery, Xihua University, Ministry of Education, Chengdu 610039, China
[2]  Key Laboratory of Fluid Machinery and Engineering, Xihua University, Chengdu 610039, China
*   Correspondence: lwj@mail.xhu.edu.cn

**Abstract:** Inlet gas void fraction (IGVF) affects the cavitation evolution in a multiphase pump and easily results in a drop of the head and efficiency when cavitation is more serious. In this paper, a numerical method was performed to qualitatively and quantitatively analyze the effect of the inlet gas void fraction on the pressure and velocity characteristics of the multiphase pump at different cavitation stages. The results show that with the increase of IGVF and the development of cavitation, the pressure in the impeller flow passage is reduced, and the pressure corresponding to the cavitation region drops sharply to the saturated vapor pressure. With the decrease of the cavitation coefficient, and due to the expulsion effect of the cavitation bubbles, the relative velocity in the cavitation region becomes larger. Because of the large pressure gradient at the end of the cavitation bubbles, the kinetic energy of the fluid is insufficient to overcome the effect of the inverse pressure gradient, resulting in a backflow vortex. Investigations on cavitation evolution in the multiphase pump at different IGVFs are of great significance for improving its performance.

**Keywords:** cavitation evolution; gas-liquid two-phase; cavitation stage; CFD; velocity distribution; backflow vortex

## 1. Introduction

With the continuous expansion of the global energy demand, the exploitation of oil and gas in human society is increasing, and multiphase transportation technology is also emerging [1–3]. As the core piece of equipment of the multiphase transportation system, the multiphase pump has the advantages of large flow, small volume and insensitivity to solid particles [4,5]. Therefore, many scientific research institutions, universities and oil companies have carried out plenty of investigations on it [6–9]. However, the inlet gas void fraction (IGVF) always changes during the operation of the pump, which gives rise to the variation of the flow field. In some cases, a low-pressure region appears, which easily leads to cavitation in the multiphase pump [10–13]. Cavitation seriously affects the operation of the multiphase pump, and it is of great engineering significance to investigate the effect of an IGVF on the flow characteristics in a multiphase pump in a cavitation case.

At present, the investigations on the flow characteristics of multiphase pumps mainly focus on the internal flow behaviors and structural optimization design, without considering the effect of cavitation. Yu Zhiyi et al. [14] employed numerical and experimental methods to investigate the gas distribution in the multiphase pump and the effect of the IGVF on the head under the gas-liquid two-phase condition. Zhang Jinya et al. [15] took water and air as the medium and conducted steady and unsteady numerical simulations of a multistage multiphase pump. When the IGVF was less than 10% and greater than 90%, the gas was regarded as an incompressible fluid; at other IGVFs, the gas was regarded as a compressible fluid. Yang Xiao qiang et al. [16] conducted an experiment on the external

characteristics of the twin-screw multiphase pump under different pressure differences and the IGVFs. The relationship between output power and IGVF and the inlet pressure difference were studied after analyzing the experimental data. Ma Xijin et al. [17] carried out a three-dimensional numerical simulation of the flow characteristics of a multiphase pump at different IGVFs, and investigated the effect of impeller blade number on multiphase pump performance. The results showed that the appropriate increase in blade number increased the pump head. Yang, X. et al. [18] adopted N32 oil and air as the medium to conduct theoretical and experimental investigations on the multiphase pump at high IGVF. The experimental data were obtained by changing the IGVF and inlet pressure. The leakage in the multiphase pump at a given pressure was increased with the increase of the IGVF, and the shaft power wasted via the high-pressure reflux reduced the pump's efficiency. Zhang, J.Y. et al. [19,20] used a multi-stage multiphase pump as the research object and adopted PIV and high-speed photography to capture the gas-liquid two-phase flow at the pump inlet. As the IGVF increased, the gas-liquid flow patterns were divided into isolated bubble flow, bubbly flow, gas pocket flow and segregated gas flow. Shi, Y. et al. [21] compared the experimental data of the three-stage multiphase pump with the numerical results under different turbulence models, and uncovered the effect of the turbulence model, wall roughness, bubble size and interphase resistance model on the multiphase pump's performance. The simulation results with the SST turbulence model were more consistent with the experimental case. Kim, J.H. et al. [22] took the multiphase pump efficiency as the objective function for optimization. By defining the diffusion-area ratio, the straight-blade length ratio, and the length ratio between the trailing edge (TE) of the impeller blade and the leading edge (LE) of the diffuser blade, a radial basis function neural network was used to optimize the multiphase pump blade. The best efficiency point of the multiphase pump was increased by 9.75%, and the multiphase pump efficiency was also improved at the large flow rate. By establishing a more appropriate and reliable numerical analysis method, Suh, J.W. et al. [23] revealed the effect of the IGVF on the flow characteristics of the multiphase pump. Zhang, W.W. et al. [24] investigated the pressure fluctuation characteristics of the multiphase pump under the gas-liquid two-phase case, and disclosed the relationship between the IGVF and the pressure fluctuation. The low IGVF had an inhibitory effect on the pressure pulsation caused by interference at a certain degree. When the IGVF reached a certain level, the amplitude of the pressure fluctuation became larger due to the enhancement of the interphase effect. In terms of the thermal effect of cavitation, Ge Mingming, et al. [25–28] studied the influence law of temperature on cavitation dynamics in a Venturi tunnel with experimental methods, divided the law of cavitation evolution, and studied the influence of temperature on the cavitation structure. The research deepened scholars' understanding of the cavitation law and has a very important significance. In terms of optimal design, Shi, G. et al. [29] used orthogonal optimization and the CFD method to optimize the gas-liquid conveying capacity of a multiphase pump. When the IGVF was 15%, the optimized pump head and efficiency were increased by 2.81% and 5.6% respectively. By considering the pump head and efficiency, Wang C et al. [30] established an optimization numerical model and used an artificial intelligence algorithm to optimize, so the pump performance had been improved. Nguyen, V et al. [31] proposed some design methods for centrifugal pumps that had great effects on the improvement of pump performance. Similarly, Zhang, W.W. et al. [32], Kim, H. et al. [33], Liu, M. et al. [34] and Peng, C. et al. [35] adopted different optimization methods to optimize the performance of the multiphase pump and improve the working capacity of the pump.

To sum up, there are currently few investigations on the flow characteristics of multiphase pumps in cavitation case. Meanwhile, the effect of IGVF on the flow characteristics of multiphase pumps under cavitation is seldom reported. Therefore, under different cavitation stages, the effect of IGVF on the flow characteristics of multiphase pumps was qualitatively and quantitatively investigated, which had important scientific significance in improving the flow stability of multiphase pumps.

## 2. Computational Model and Method of Multiphase Pump

### 2.1. Computational Model

An impeller was selected to investigate the cavitation characteristics of a multiphase pump in this paper, and the computational domain was composed of inlet pipe, impeller pipe, and outlet pipe. To achieve full flow at the impeller inlet and outlet, the inlet and outlet were respectively extended to 2 and 6 times the axial length of the impeller, as shown in Figure 1. The design parameters of the pump are as follows: design flow rate Q = 100 m$^3$·h$^{-1}$, speed n = 3000 rpm, impeller blade number Z = 4, impeller blade wrap angle 179.6°, impeller inlet hub ratio 0.79, impeller outlet hub ratio 0.74, and impeller diameter 230.5 mm.

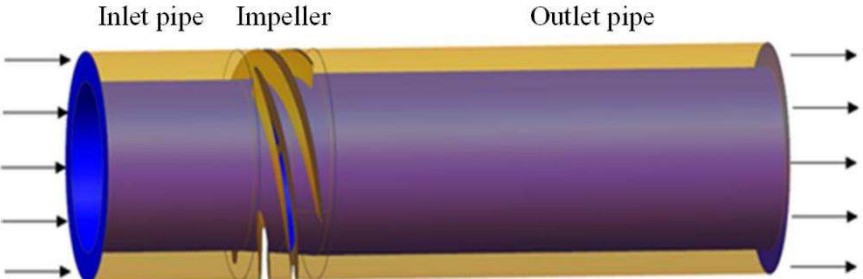

**Figure 1.** Three-dimension model of multiphase pump.

### 2.2. Numerical Simulation Theory

2.2.1. Turbulence Model

The multiphase pump blade has a large curvature and high speed, and flow separation occurs in the multiphase media case. Therefore, the SST *k-ω* turbulence model was selected in this paper.

The *k*-epsilon turbulence model has weak results in predicting strongly separated flows, flows containing a large curvature and flows with a strong pressure gradient. The SST *k-ω* turbulence model combines the advantages of the *k-ω* model and the *k-ε* model, which better deals with the viscous flow in the near-wall region and the turbulent flow in the far field. The expressions of the turbulent kinetic energy *k* and the turbulent pulsation frequency w of the SST *k-ω* model are as follows:

$$\rho \frac{\partial k}{\partial t} + \rho v_j \frac{\partial k}{\partial x_j} = \frac{\partial}{\partial x_j}\left[\left(u + \frac{u_t}{\sigma_k}\right)\frac{\partial k}{\partial x_j}\right] + u_t \frac{\partial v_j}{\partial x_i}\left(\frac{\partial v_j}{\partial x_i} + \frac{\partial v_i}{\partial x_j}\right) - \beta^* \rho k w \tag{1}$$

$$\rho \frac{\partial w}{\partial t} + \rho v_j \frac{\partial w}{\partial x_j} = \frac{\partial}{\partial x_j}\left[\left(u + \frac{u_t}{\sigma_w}\right)\frac{\partial w}{\partial x_j}\right] + \frac{\alpha w}{k} u_t \frac{\partial v_j}{\partial x_i}\left(\frac{\partial v_j}{\partial x_i} + \frac{\partial v_i}{\partial x_j}\right) - \beta \rho k w^2 + 2(1 - F_1)\rho \frac{1}{w\sigma_w}\frac{\partial k}{\partial x_j}\frac{\partial w}{\partial x_j} \tag{2}$$

where $\rho$ is the density, $\beta$, $\beta^*$ and $\sigma_w$ are the empirical coefficient, and $F_1$ is the blending function.

2.2.2. Multiphase Flow Model

In this paper, the range of IGVF in the multiphase pump was 0~20%, and the volume fraction of bubbles varied greatly and its distribution range was wide. The cavitation flow was more complicated, and the interface between the vapor and liquid phases was not clearly defined. Considering the computational cost, the mixture model was selected to perform the numerical simulation of the multiphase flow in the multiphase pump.

The mixture model is used commonly for multiphase flow in engineering, which allows two phases to intersect each other. The volume fraction of the two phases in a control body can be any value between 0 and 1, and there is a speed slip between the two phases. The expressions of the continuity equation, momentum equation, energy equation,

the volume fraction equation of the second phase and the relative velocity in the mixture model are as follows:

$$\frac{\partial}{\partial t}(\rho_m) + \nabla \cdot \left(\rho_m \vec{V}_m\right) = 0 \tag{3}$$

$$\frac{\partial}{\partial t}\left(\rho_m \vec{V}_m\right) + \nabla \cdot \left(\rho_m \vec{V}_m \vec{V}_m\right) = -\nabla p + \nabla \cdot \left[\mu_m\left(\nabla \vec{V}_m + \nabla \vec{V}_m^T\right) + \rho_m \vec{g} + \vec{F} + \nabla \cdot \left(\sum_{k=1}^{n} \alpha_k \rho_k \vec{V}_{dr,k} \vec{V}_{dr,k}\right)\right] \tag{4}$$

$$\frac{\partial}{\partial t}\sum_{k=1}^{n}(\alpha_k \rho_k E_k) + \nabla \cdot \sum_{k=1}^{n}\left[\alpha_k \vec{V}_k(\rho_k E_k + \rho)\right] = \nabla \cdot \left(k_{eff}\nabla T\right) + S_E \tag{5}$$

$$\frac{\partial}{\partial t}(\alpha_p \rho_p) + \nabla \cdot \left(\alpha_p \rho_p \vec{V}_m\right) = -\nabla \cdot \left(\alpha_p \rho_p \vec{V}_{dr,p}\right) \tag{6}$$

$$\mu_m = \sum_{k=1}^{n} \alpha_k \mu_k \tag{7}$$

$$\sum_{k=1}^{n} \alpha_k = 1 \tag{8}$$

$$\rho_m = \sum_{k=1}^{n} \alpha_k \rho_k \tag{9}$$

$$\vec{V}_m = \frac{\sum_{k=1}^{n} \alpha_k \rho_k \vec{V}_k}{\rho_m} \tag{10}$$

Drift velocity of the *k* phase is

$$\vec{V}_{dr,k} = \vec{V}_k - \vec{V}_m \tag{11}$$

where $\rho_m$ and $\rho_k$ are the density of the mixture and *k* phase; $\vec{V}_{dr,k}$, $\vec{V}_k$ and $\vec{V}_m$ are the drift velocity of the *k* phase, the velocity of the *k* phase and the average velocity of the mass; $\alpha_k$ is the volume fraction of the *k* phase; *p* is the pressure; $\mu_m$ and $\mu_k$ are the dynamic viscosity of the mixture phase and *k* phase; $\vec{g}$ is the acceleration of gravity; $\vec{F}$ is body force; $k_{eff}$ is the effective thermal conductivity; $E_k$ is the energy of the *k* phase; *T* is the temperature; and $S_E$ includes the contribution of all other volumetric heat sources.

The mixture model is a heterogeneous model that uses slip velocity to allow for a different velocity between phases. The slip velocity is also the relative velocity, which refers to the velocity of the secondary phase (*p*) with respect to the primary phase (*q*)

$$\vec{V}_{qp} = \vec{V}_p - \vec{V}_q \tag{12}$$

The slip velocity is expressed as follows:

$$\vec{V}_{qp} = \frac{\tau_p}{f_{drag}}\frac{\rho_p - \rho_m}{\rho_p}\vec{a} \tag{13}$$

where $\vec{a}$ is the acceleration of the secondary phase (gas or vapor in cavitation); $\tau_p$ is the relaxation time of the ***p*** phase; and $f_{drag}$ is the drag function.

$$\vec{a} = \vec{g} - (\vec{V}_m \cdot \nabla)\vec{V}_m - \frac{\partial \vec{V}_m}{\partial t} \tag{14}$$

$$\tau_p = \frac{\rho_p d_p^2}{18\mu_q} \tag{15}$$

where $d_p$ is the $p$ phase particle diameter.

### 2.2.3. Cavitation Model

The simulation results based on the Singhal cavitation model have a large deviation from the experimental data and over-predict the range of cavitation. The Schnerr–Sauer cavitation model predicts that the evolution period of the cavitation flow is smaller than the experimental value, and cannot accurately predict the evolution and development of cavitation. The Zwart–Gerber–Belamri cavitation model accurately simulates the quasi-periodical and evolution process of cavitation. Therefore, the Zwart–Gerber–Belamri cavitation model was used to simulate the cavitation flow in the multiphase pump. The interphase transmission rate of this model is as follows:

when $P \leq P_v$

$$R_e = F_{vap} \frac{3\alpha_{nuc}(1-\alpha_v)\rho_v}{R_B} \sqrt{\frac{2(P_v - P)}{3\rho_l}} \tag{16}$$

when $P > P_v$

$$R_c = F_{cond} \frac{3\alpha_v\rho_v}{R_B} \sqrt{\frac{2(P - P_v)}{3\rho_l}} \tag{17}$$

In the formula, $R_B$ is the bubble radius, $10^{-6}$ m; $\alpha_{nuc}$ is the volume fraction of the vapor nucleus position, $5 \times 10^{-4}$; $F_{vap}$ and $F_{cond}$ represent the vapor evaporation and condensation coefficients, and the values are 50 and 0.01, respectively.

### 2.3. Mesh and Independent Verification

The structural hexahedral mesh was adopted on the single-flow passage of the pump impeller, and ICEM CFD software was performed to rotate and copy the single-flow passage mesh into a full-flow passage. Meanwhile, the ICEM software was also used to arrange structural hexahedral mesh on the inlet and outlet extensions. The advantage of structural hexahedral mesh is that the mesh nodes can be adjusted to optimize the local mesh, and the flow details can be clearly presented. The meshes of the inlet pipe, outlet pipe and impeller are shown in Figure 2, and the mesh quality is shown in Table 1.

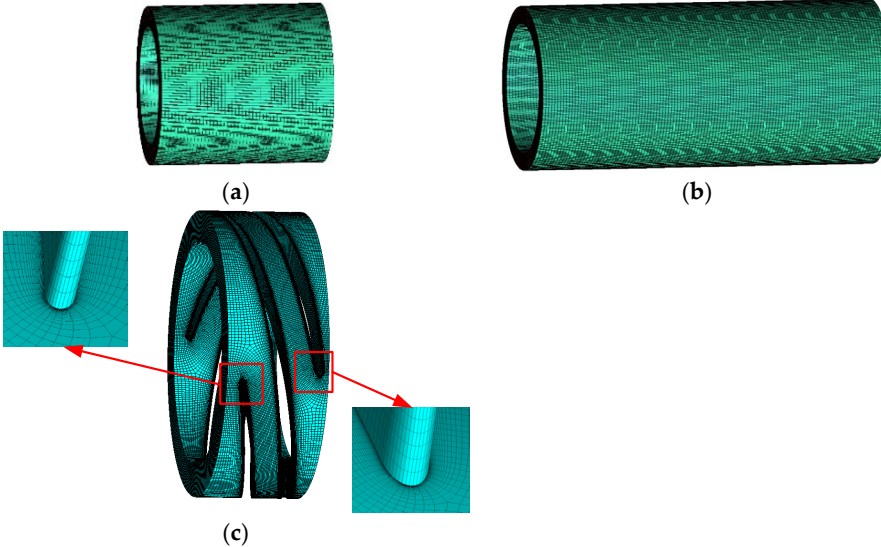

**Figure 2.** Computational mesh (**a**) inlet pipe; (**b**) outlet pipe; and (**c**) impeller.

**Table 1.** Mesh quality.

| Computational Domain | Max Angle | Max Warp | Skew | Aspect Ratio | Quality |
|---|---|---|---|---|---|
| Inlet pipe | 90.0377~108.284 | 0~0.416 | 0.80~0.99 | 0.00091~0.99 | 0.95 |
| Impeller | 90.002~121.61 | 0~2.46 | 0.68~1 | 0.0018~0.99 | 0.48 |
| Outlet pipe | 90.001~119.16 | 0~0.27 | 0.688~1 | 0.0006~0.99 | 0.87 |

To improve the accuracy and efficiency of the numerical simulation, the mesh independence was verified. A total of 7 sets of meshes on the multiphase pump were performed and simulated at water case and design flow rate. The appropriate mesh number was selected for the final numerical simulation. The mesh independent verification is shown in Figure 3.

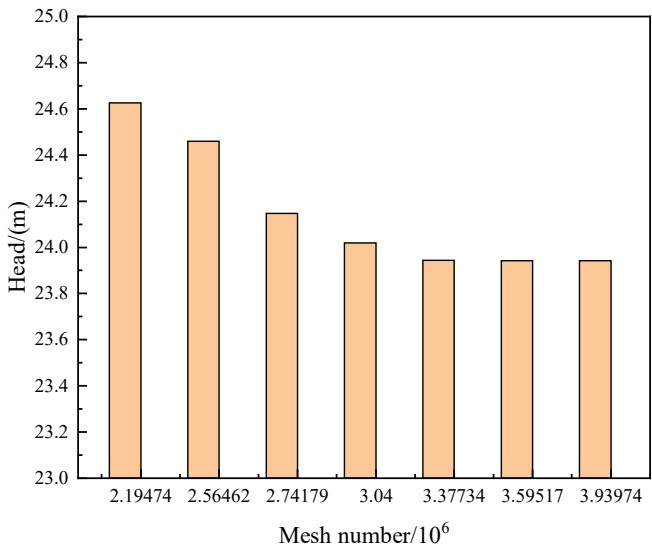

**Figure 3.** Mesh independent verification.

From Figure 3, as the mesh number increases, the head gradually decreases and stabilizes. When the mesh number is greater than 3.38 million, the head changes only by 0.31%, which is less than 0.5%. Therefore, 3.38 million mesh was more appropriate in the final simulation. The meshes of the inlet pipe, impeller and outlet pipe were 500,000, 2,170,000 and 900,000, respectively.

### 2.4. Boundary Condition Settings

ANSYS Fluent software was used to simulate the steady cavitation flow in the multiphase pump at the design case. The turbulence model, multiphase flow model and cavitation model were employed as described above. The inlet and outlet boundaries were set to pressure inlet and mass flow, respectively. The IGVF was set to 0 at the inlet, and the saturated vapor pressure of water at 25 °C was 3170 Pa. Cavitation occurs in the pump by gradually reducing the pressure at the multiphase pump inlet. The interfaces between the impeller and the inlet pipe, the impeller and outlet pipe are set as the interface. The impeller hub, blade and shroud are set as relatively no-slip walls, and the other walls are absolutely no-slip walls. The convergence accuracy is set to $10^{-5}$.

## 3. Experimental Rig and Numerical Verification

### 3.1. Experimental Rig

The experimental rig of the multiphase pump included the motor, multiphase pump, gas-liquid mixing tank, lubrication system, cooling system, control system, water supply

system, gas supply system, test system, pipeline and valves, etc. The experimental system for the multiphase pump is shown in Figure 4.

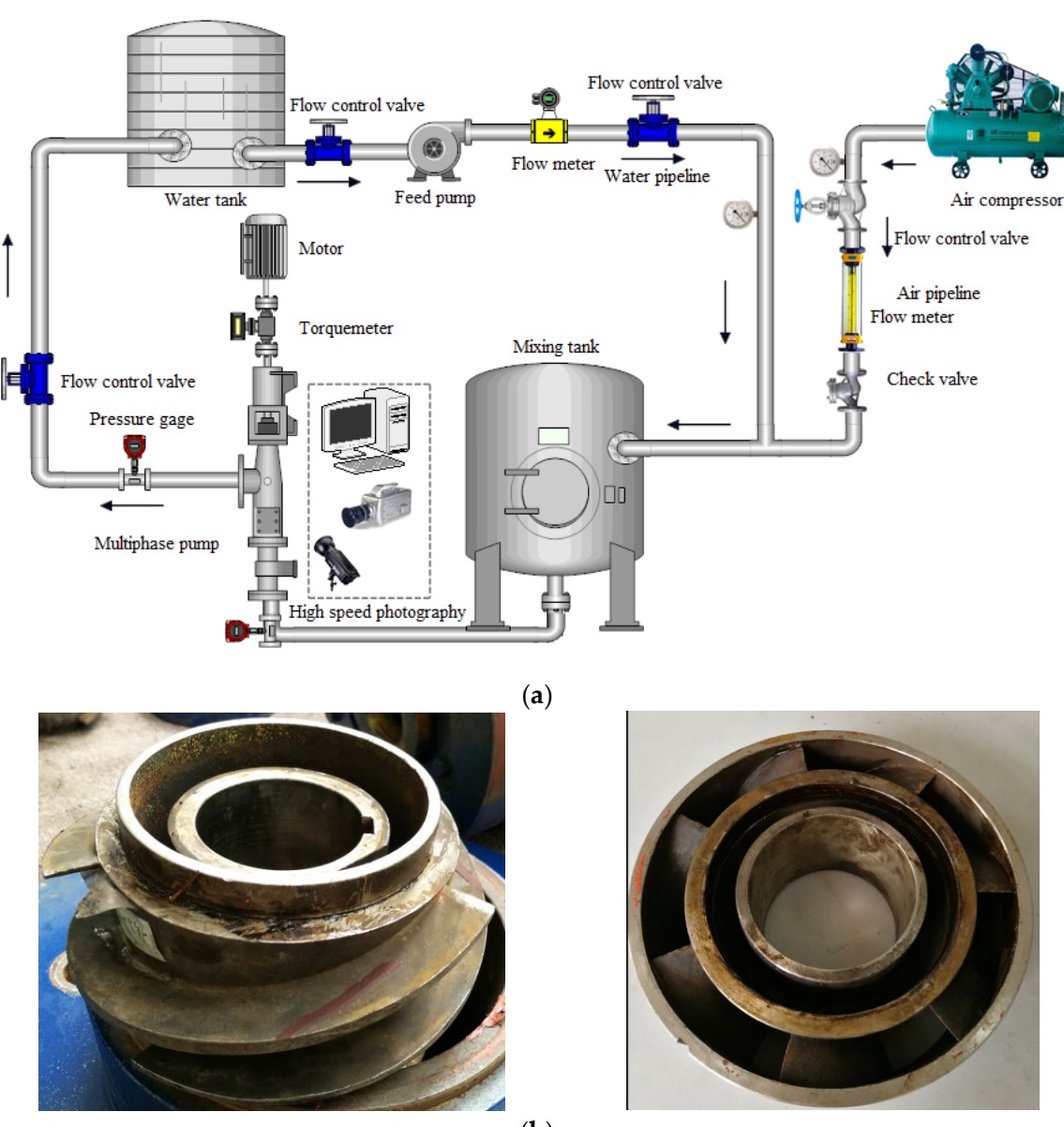

(**a**)

(**b**)

**Figure 4.** Experimental rig of the multiphase pump. (**a**) Schematic diagram. (**b**) Impeller and diffuser.

In the experiment, the inlet and outlet pressure gauges were used to measure the inlet and outlet pressure of the multiphase pump. A torque meter was used to measure speed, power and torque. Repeated tests were carried out for each operating point to prevent errors in the test process.

### 3.2. Numerical Verification

Figure 5 shows the comparison between the CFD and the experiment, and the head and efficiency in the CFD agreed well with the experimental one. Meanwhile, the relative errors of the head and efficiency at the optimal point were 4.1% and 4.1%, respectively. The maximum relative errors of the head and the efficiency were both no more than 5%, and this shows that the simulation is reliable.

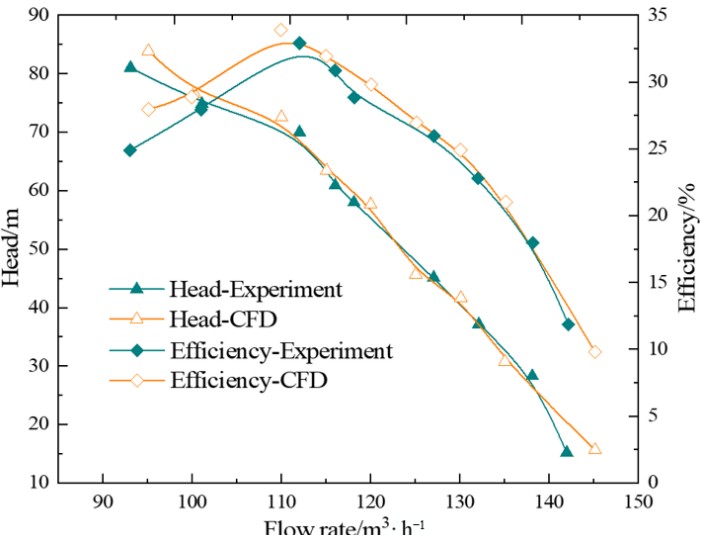

**Figure 5.** Hydraulic characteristics between CFD and experiment.

## 4. Results and Discussion

*4.1. Cavitation Characteristic Prediction*

The head coefficient $\psi$ and cavitation coefficient $\sigma$ are shown as follows:

$$\psi = \frac{P_{out} - P_{in}}{0.5\rho U^2} \tag{18}$$

$$\sigma = \frac{P_{in} - P_v}{\rho U^2/2} \tag{19}$$

where $P_{in}$ and $P_{out}$ denote the multiphase pump inlet and outlet pressure, respectively. $U$ is the circumferential velocity of the impeller hub, expressed as $U = \pi Dn/60$. $P_v$ is the saturated vapor pressure of water at 25 °C, 3170 Pa.

Figure 6 shows that the head coefficient $\psi$ of the multiphase pump varies with the cavitation coefficient $\sigma$. For Figure 6, under IGVF = 0, 0.1 and 0.2, with the decrease of the cavitation coefficient $\sigma$, the head coefficient $\psi$ remains unchanged at first, then steadily decreases, and finally decreases sharply.

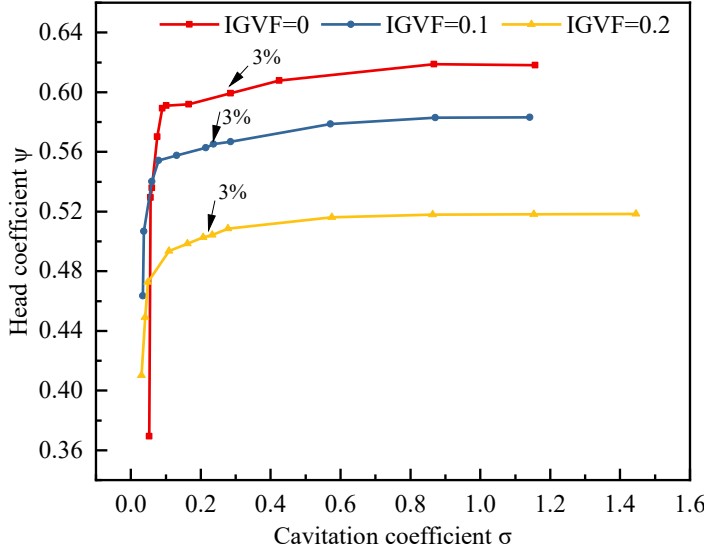

**Figure 6.** Cavitation characteristics.

Under IGVF = 0, when the cavitation coefficient $\sigma$ is greater than 0.86, the head coefficient $\psi$ remains unchanged, indicating that there is no cavitation, or the cavitation in the multiphase pump is relatively weak and does not affect the head. When the cavitation coefficient $\sigma$ is between 0.106~0.86, the head coefficient $\psi$ gradually decreases. When the cavitation coefficient $\sigma$ is 0.28, the head coefficient $\psi$ is reduced by 3% compared to the no-cavitation case, which is called critical cavitation in engineering. When the cavitation coefficient $\sigma$ is less than 0.106, the head coefficient $\psi$ decreases sharply. When $\sigma$ decreases to 0.077, the head coefficient decreases by 7.68%, which indicates that the degree of cavitation in the multiphase pump is more serious. When the cavitation coefficient $\sigma$ is 0.051, the head reduction is exceeded by 20%, and the cavitation reaches the fractured cavitation.

In the case of IGVF = 0.1, the cavitation coefficient $\sigma$ is greater than 0.86, and the head coefficient $\psi$ remains unchanged. When the cavitation coefficient $\sigma$ gradually decreases between 0.0769 and 0.86, the head coefficient $\psi$ decreases steadily, and the head coefficient $\psi$ corresponding to the critical cavitation is 0.24. When the cavitation coefficient $\sigma$ is less than 0.0769, the head coefficient $\psi$ drops sharply. When the head coefficient $\psi$ decreases by 7.3%, the cavitation coefficient $\sigma$ is 0.057. The cavitation coefficient $\sigma$ corresponding to a decrease of more than 20% in the head coefficient $\psi$ is 0.033.

Under IGVF = 0.2, the head coefficient $\psi$ remains unchanged in the range of cavitation coefficient $\sigma$ greater than 0.86. When the cavitation coefficient $\sigma$ gradually decreases in the range of 0.107~0.86, the head coefficient $\psi$ of the multiphase pump is reduced steadily, and the critical cavitation coefficient $\sigma$ is 0.208. When the cavitation coefficient $\sigma$ gradually decreases in the range of less than 0.107, the head coefficient $\psi$ decreases greatly. When the cavitation coefficient $\sigma$ is 0.048, the head coefficient $\psi$ decreases by 7.4%; when the cavitation coefficient $\sigma$ is 0.029, the head coefficient $\psi$ decreases by more than 20%.

*4.2. Effect of IGVF on Pressure Characteristics at Cavitation Case*

To analyze the effect of the cavitation development on the pressure characteristics in the multiphase pump impeller, the pressure was analyzed under the different IGVFs and cavitation states. Figure 7 shows that the pressure in the impeller passage at 0.1, 0.5, and 0.9 span IGVF = 0, 0.1, and 0.2 under different cavitation stages.

Under different operating points, because the impeller blade works on the fluid, the pressure in the flow passage gradually increases from the impeller inlet to the outlet; as the span increases, the pressure gradually rises. In the critical cavitation case, with the increase of the IGVF, the pressure in the impeller flow passage at the same span gradually decreases; that is, the increase of the IGVF reduces the pressurization performance of the multiphase pump. When the cavitation develops to the second stage, the pressure in the impeller flow passage becomes smaller than that during critical cavitation, and the pressure drop corresponding to the cavitation region is larger. When the cavitation develops to the third stage, the degree of cavitation is very serious, and cavitation appears at the suction side (SS) and pressure side (PS). The pressure in the cavitation region is very small. In the IGVF = 0 case, the cavitation occupies the entire blade SS, and the pressure in this region is almost 0.

To explore the effect of cavitation development on the pressure on the blade surface, the pressure on the 0.5 blade span was analyzed. Figure 8 displays the pressure along the streamwise at the 0.5 blade span under different cavitation stages. The pressure on the PS is higher, and the pressure on the SS is lower. The area enclosed by the pressure between the PS and SS can reflect the blade load.

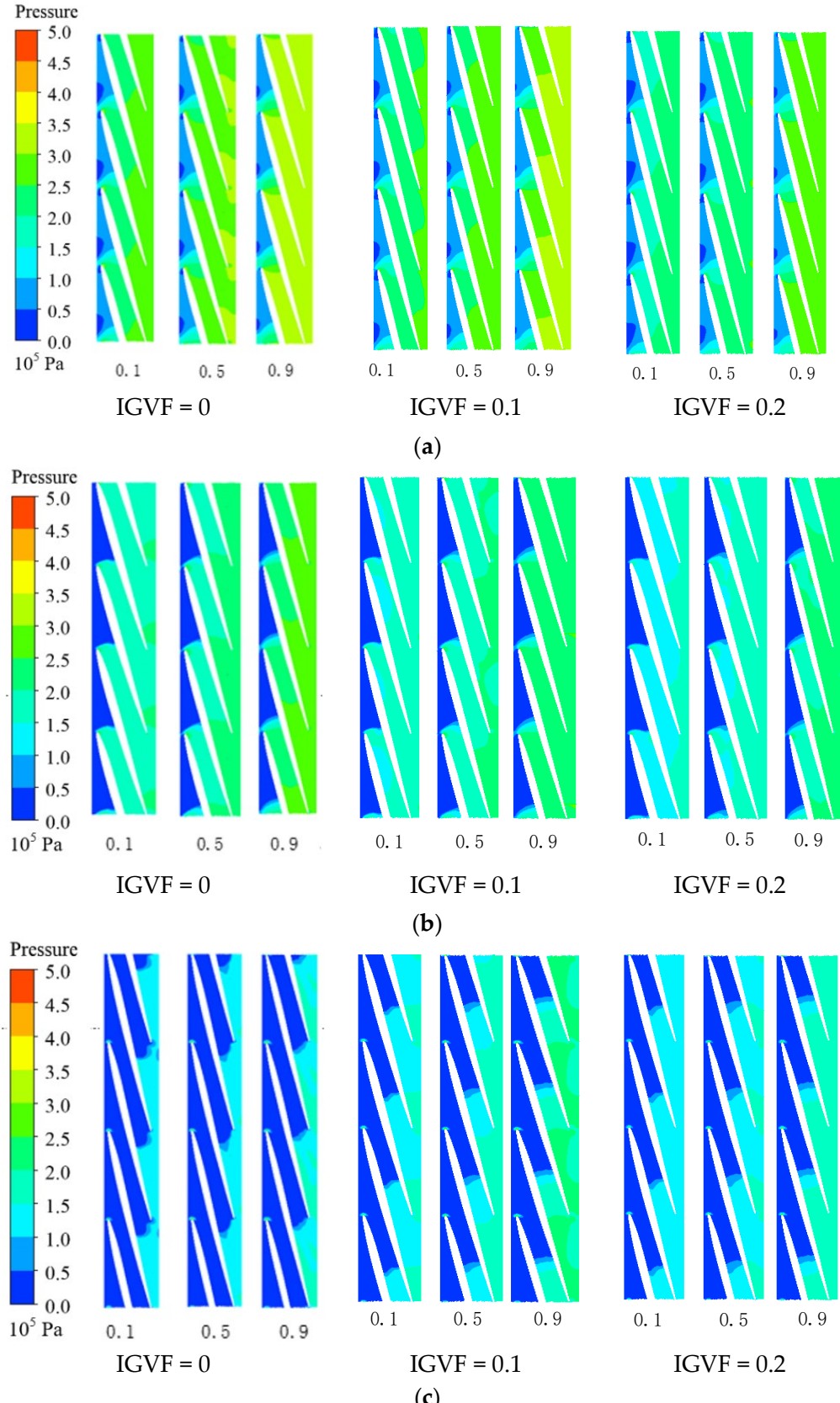

**Figure 7.** Pressure in impeller flow passage at IGVF = 0, 0.1, 0.2. (**a**) Critical cavitation; (**b**) severe cavitation; (**c**) fracture cavitation.

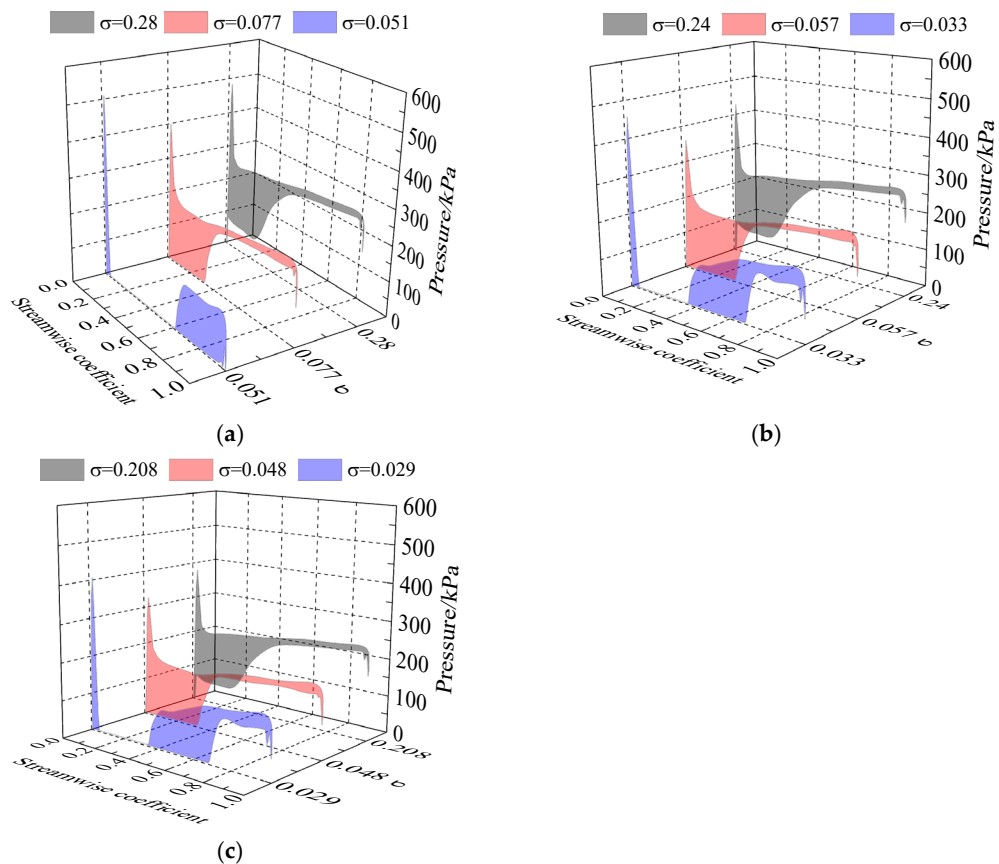

**Figure 8.** Pressure on 0.5 blade span at IGVF = 0, 0.1, 0.2. (**a**) IGVF = 0; (**b**) IGVF = 0.1; (**c**) IGVF = 0.2.

In Figure 8a, when the cavitation is weak, the pressure on the PS increases steadily from the inlet to the outlet along the streamwise. However, the pressure on the SS first maintains 80 kPa in the streamwise of 0~0.3, then increases, and finally remains at a steady increasing trend. When the cavitation coefficient is reduced to 0.077, the pressure on the blade SS is basically unchanged from the critical cavitation. Meanwhile, the pressure on the blade SS in the streamwise of 0~0.32 is about 3170 Pa. When the cavitation coefficient $\sigma$ is reduced to 0.051, the cavitation on the blade PS extends to the streamwise of 0.05~0.62, and the SS is completely occupied by cavitation. Then, the blade load in the streamwise of 0.05~0.62 becomes 0, and the blade load from the streamwise of 0.62 to the blade outlet becomes larger.

From Figure 8b,c, with the increase of IGVF, in the critical cavitation the pressure on the blade PS at the same location gradually decreases, and the blade load gradually decreases. When the cavitation is in the second stage—that is, $\sigma$ = 0.057 and 0.048—since cavitation occurs in the streamwise of 0~0.35 on the blade SS, the pressure on the blade SS is reduced to about the saturated vapor pressure. When the cavitation coefficient $\sigma$ is reduced to 0.033 and 0.029, the cavitation appears in the streamwise of 0.05~0.38 and 0~0.7 on the blade PS and SS, and the pressure on the blade in this range is reduced to about 3170 Pa. The blade load in the cavitation intersects between the PS and SS (in the streamwise of 0.05~0.38) decreases to 0. However, the blade load in the streamwise of 0.38~0.7 gradually increases. In addition, in the second and third stages of cavitation at different IGVFs, the pressure gradient corresponding to the end of the cavitation on the blade surface is relatively large.

Based on the above analysis, the increase of the IGVF makes the pressure in the impeller flow passage decrease, and the blade load decreases. The pressure corresponding to the bubble region is reduced to about the saturated vapor pressure.

### 4.3. Effect of IGVF on Velocity Characteristics at Cavitation Case

To analyze the effect of cavitation evolution on the flow pattern in the impeller, the velocity in the impeller passage at different spans is analyzed below. Figure 9 displays the relative velocity in the impeller at different spans with IGVF of 0, 0.1, and 0.2 under different cavitation states.

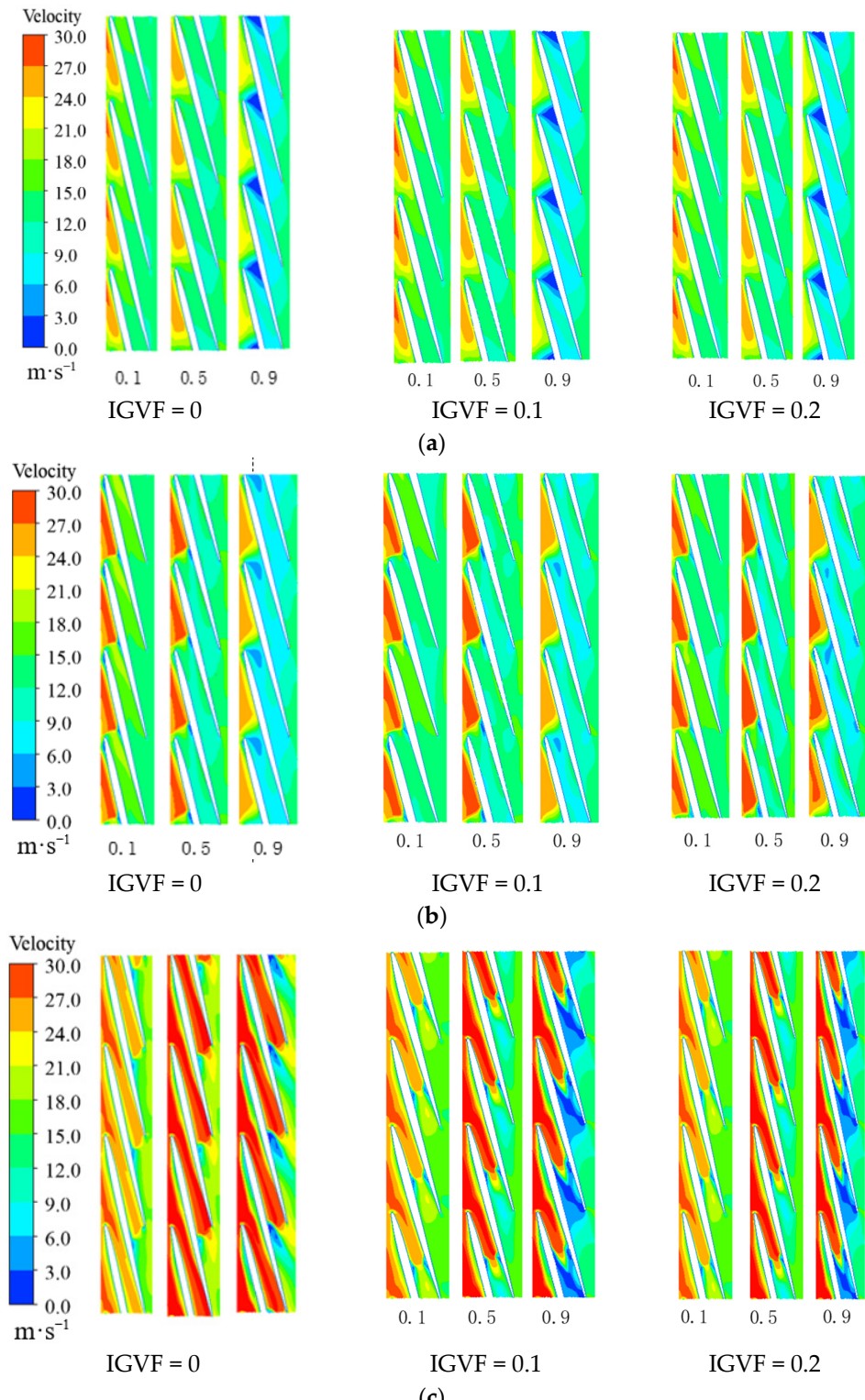

**Figure 9.** Velocity in impeller under IGVF = 0, 0.1 and 0.2. (**a**) Critical cavitation; (**b**) severe cavitation; and (**c**) fracture cavitation.

In Figure 9, in the critical cavitation at different IGVFs, the kinetic energy and pressure energy of the fluid are mutually converted during the fluid flows from the impeller inlet to the outlet, resulting in a gradual decrease in velocity. The relative velocity also gradually decreases from the hub to the shroud. When the cavitation develops to the second stage, the relative velocity in the cavitation region becomes larger, and as the span increases, the relative velocity gradually decreases. When the cavitation develops to the third stage, the cavitation in the impeller is more serious, and the cavitation range increases. The relative velocity in the cavitation region increases more so than in the previous two cavitation stages. Compared to the first two cavitation stages, as the blade span increases, the relative velocity in the cavitation region gradually increases; this indicates that the flow state in the impeller changes greatly, and the velocity distribution from the end of the cavitation to the outlet is more turbulent when the cavitation is severe. In addition, there is a low-velocity region at the end of the cavitation zone. From the pressure distribution, this is because the pressure gradient at the end of the cavitation region is large, which results in a larger velocity reduction.

To quantitatively analyze the effect of the development of cavitation on the velocity distribution under the different IGVFs, the velocity distribution at 0.5 span along the streamline direction is analyzed below. Figure 10 shows the velocity distribution at 0.5 span along the streamwise with IGVF of 0, 0.1 and 0.2 under different cavitation stages.

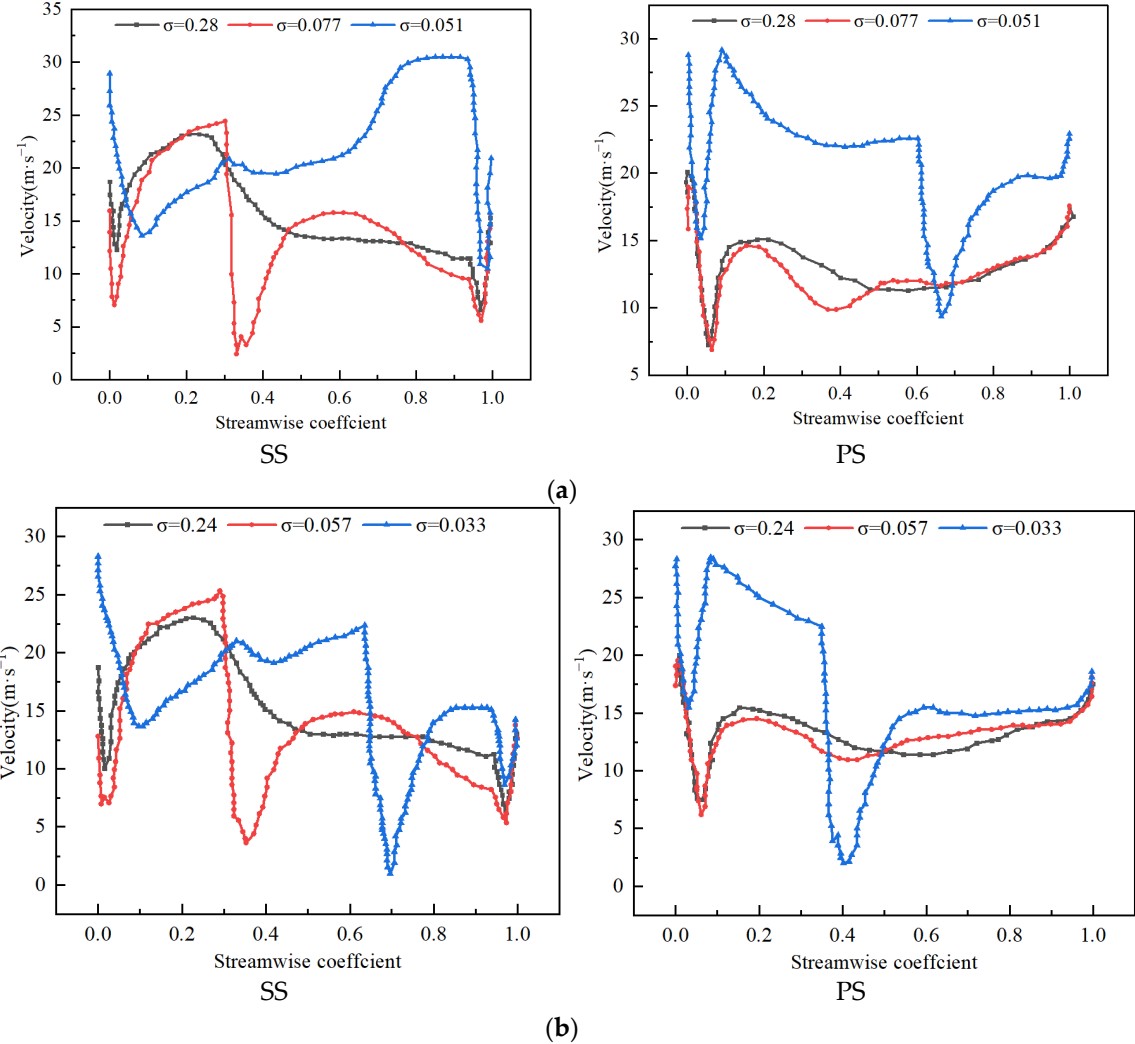

**Figure 10.** *Cont.*

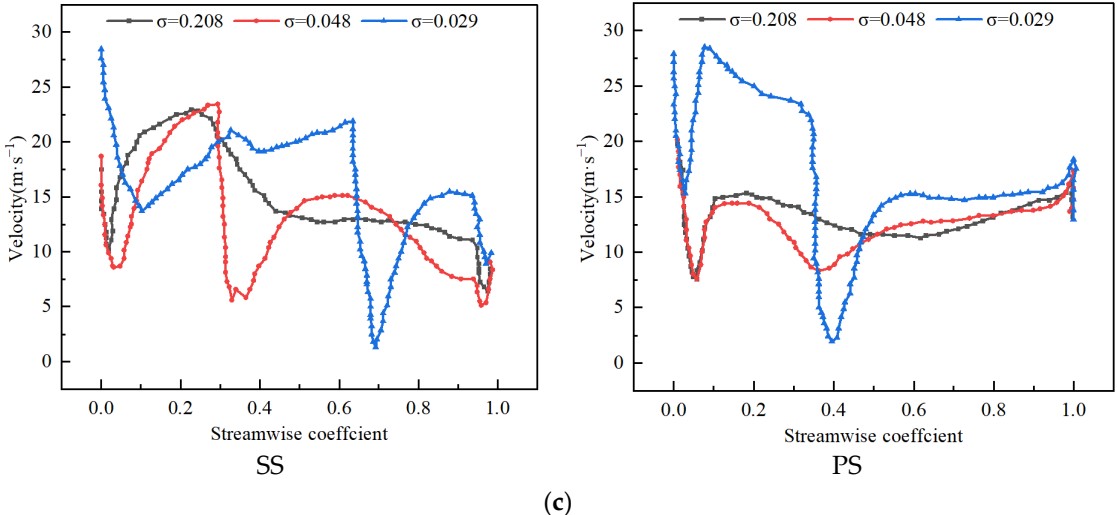

(**c**)

**Figure 10.** Velocity distribution at 0.5 span under IGVF = 0, 0.1, 0.2. (**a**) IGVF = 0; (**b**) IGVF = 0.1; (**c**) IGVF = 0.2.

From Figure 10, at different IGVFs, when cavitation develops to the second stage, compared to the critical cavitation, the relative velocity gradient at blade SS becomes larger in the streamwise of 0~0.1. From the previous distribution of bubbles, in this cavitation stage, the bubbles are attached to the blade SS, which thickens the boundary layer around the inlet of the blade SS and results in a large change in the velocity gradient. However, in the streamwise of 0.3 to 0.4, the velocity gradient changes greatly compared to the critical cavitation, which is caused by the larger pressure gradient at the end of the bubbles on the blade surface. When the cavitation develops to the third stage, at IGVF = 0 the cavitation is very serious, and the cavitation occupies the entire blade SS and most of the impeller passage; this leads to an increase in the velocity on the blade SS from the streamwise of 0.3 to the outlet. For IGVF = 0.1 and 0.2, because the cavitation phenomenon extends along the blade SS to the streamwise of 0.7, the pressure gradient in the streamwise of 0.6 to 0.8 becomes larger, resulting in a larger change in the velocity gradient. Under different IGVFs, the large velocity gradient on the PS is also caused by the large pressure gradient at the end of the bubbles. Due to the squeezing effect of the bubbles, the relative velocity outside of the bubble area increases.

To observe the flow pattern in the impeller passage more intuitively, the velocity streamlines at 0.5 span are used to represent the fluid behaviors. Figure 11 is the velocity streamlines at 0.5 span with IGVF = 0, 0.1 and 0.2 under different cavitation states.

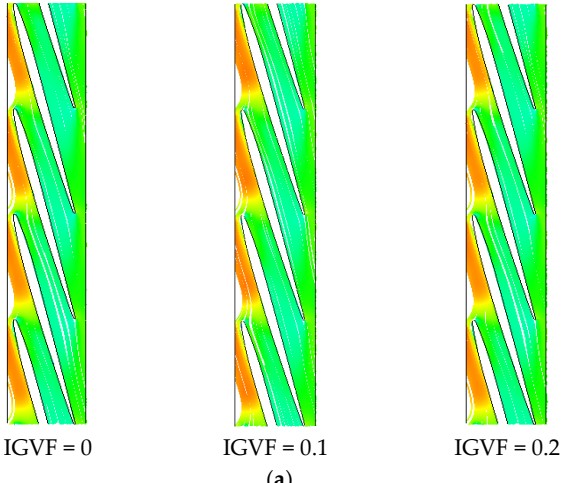

IGVF = 0        IGVF = 0.1        IGVF = 0.2

(**a**)

**Figure 11.** *Cont.*

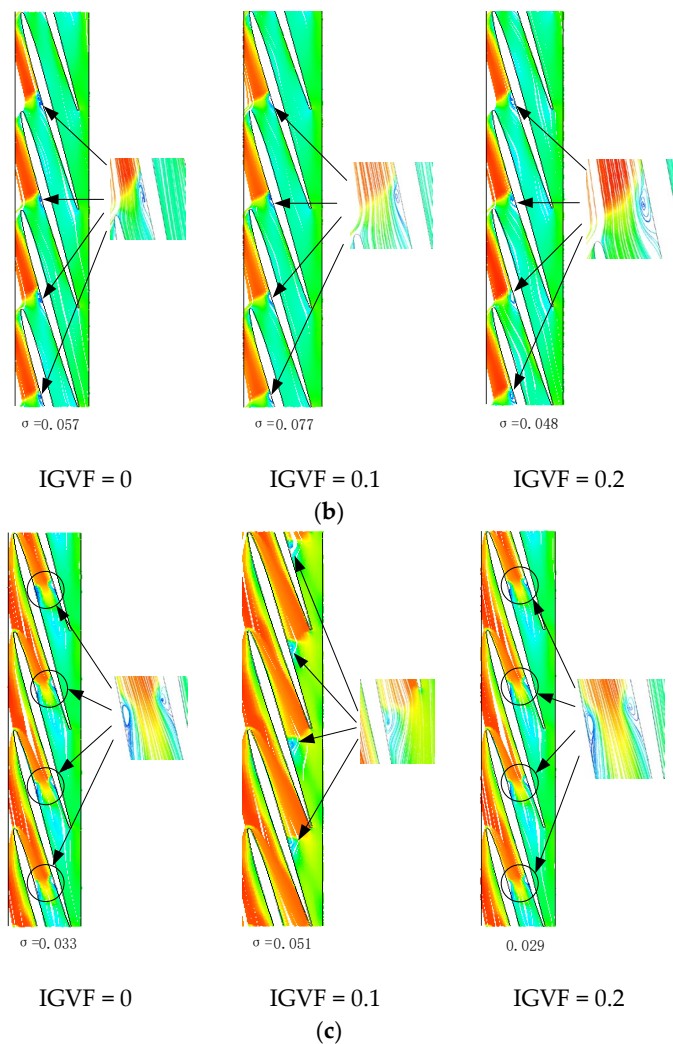

**Figure 11.** Velocity streamlines at 0.5 span under IGVF = 0, 0.1 and 0.2. (**a**) Critical cavitation; (**b**) severe cavitation; and (**c**) fracture cavitation.

From Figure 11, in the critical cavitation, the velocity streamlines in the flow passage are relatively smooth at different IGVFs. When the cavitation develops to the second stage, due to the large pressure gradient at the end of the bubbles, a backflow vortex appears at the end of the bubbles on the blade SS. When the cavitation develops to the third stage, at IGVF = 0 the backflow vortex appears at the end of the bubbles on the blade PS due to the large reverse pressure gradient. However, a backflow vortex appears at the end of the bubbles on the PS and SS with IGVF = 0.1 and 0.2.

## 5. Conclusions

The effect of cavitation flow in the multiphase pump was investigated through the analysis of the pressure, velocity and turbulent kinetic energy characteristics. The main conclusions are as follows:

(1) The increase of IGVF and the development of cavitation reduce the pressure in the impeller flow passage, and the pressure corresponding to the bubbles drops sharply to about the saturated vapor pressure. When the bubbles extend to the blade PS, the load near the cavitation intersection area is reduced to 0. The load on the end of the bubbles between the PS and SS increases, and the pressure gradient at the end of the bubbles is very large.

(2)  As the cavitation coefficient decreases, the relative velocity near the cavitation becomes larger due to the squeezing effect of the bubbles. Due to the large pressure gradient at the end of the bubbles, the kinetic energy of the fluid is not enough to overcome the effect of the reverse pressure gradient, resulting in a backflow vortex.

(3)  As the blade span increases, the relative velocity in the cavitation region gradually increases. Because the pressure gradient at the end of the cavitation region is large, a low-velocity region occurs at the end of the cavitation zone. To further avoid the cavitation phenomenon, a convex structure can be set on the blade to improve the flow state in the pump.

**Author Contributions:** Conceptualization, W.L. and H.W.; methodology, W.L.; software, W.L., H.W. and G.S.; writing—original draft preparation, H.W.; writing—review and editing, W.L. and H.W.; supervision, S.W. All authors have read and agreed to the published version of the manuscript.

**Funding:** This work was supported by the Open Research Fund Program of the State Key Laboratory of Hydroscience and Engineering (sklhse-2021-E-03, sklhse-2022-KY-06); the Key scientific research fund of Xihua University of China (Z1510417); the Central Leading Place Scientific and Technological Development Funds for Surface Project (2021ZYD0038); the National Key Research and Development Program (2018YFB0905200); and The National Natural Science Foundation of China (52279088).

**Institutional Review Board Statement:** Not applicable.

**Informed Consent Statement:** Not applicable.

**Data Availability Statement:** Not applicable.

**Conflicts of Interest:** The authors declare that they have no conflict of interest.

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
