# Peer review of "Analyzing the Inlet Gas Void Fraction on the Flow Characteristics for a Multiphase Pump Based on Cavitation Cases"

_jmse, doi:10.3390/jmse11010130_

Round 1
Reviewer 1 Report
Comments and Suggestions for Authors
The manuscript investigated the effect of cavitation flow in the multiphase pump through the analysis of the pressure, velocity, and turbulent kinetic energy characteristics. The effect of IGVF on the flow characteristics of the multiphase pump is qualitatively and quantitatively investigated under different cavitation stages. Results show that the increase of IGVF and the development of cavitation reduce the pressure in the impeller flow passage.
The work done by the author provides suggestions to improve the flow stability of multiphase pumps at different IGVFs, and has certain academic research and engineering practical value. I consider the content of this manuscript will definitely meet the reading interests of the readers of the JMSE journal. However, there are certain English spelling and grammar issues, and also the discussion and explanation should be further improved.
Therefore, I suggest giving a minor revision and the authors need to clarify some issues or supply some more experimental data to enrich the content. This could be a comprehensive and meaningful work after revision.
Detailed comments can be found in the PDF file.

Author Response
Response to Reviewer 1 Comments
Point 1:For writing issues, it is suggested that the author double-check the small grammar orinaccurate“wording”issues in the full text.
Response 1: Thanks for your helpful comments from the reviewers. I have checked the full text in detail and revised the grammar and wording. The changes are shown in red in the text.
Point 2: A lot of“Error! Reference source not found" need to be revised.
Response 2: Thanks for your helpful comments from the reviewers. I have checked the full text in detail and revised the grammar and wording. The changes are shown in red in the text.
Point 3:Line 114, for the mixture model, please put more details on the“speed slip between twophases". Provide the model or expressions on how to calculate the slip velocity in cavitation flow.
Response 3: The mixed model is a heterogeneous model which uses slip velocity to allow for different velocity between phases.The slip velocity is also the relative velocity, which refers to the velocity of the secondary phase (p) with respect to the primary phase(q)
The slip velocity is expressed as follows
where is the acceleration of the secondary phase(gas or vapor in cavitation)ï¼›is the relaxation time of p phaseï¼› is the drag function.
where is the p phase particle diameter.
The relevant changes have been added to the original text and highlighted in red.
Point 4:In lines 117 - 129, all the symbols and abbreviations should be explained at the first appearance in the manuscript, such as Vdrk.
Response 4: Thanks for the reviewer's suggestion. I have checked the formula of lines 117-129 and modified it. The modified part has been added in the original text and marked in red.
Point 5:In lines 131-134, the authors discussed the performance of models, like Singhal,Schnerr-Sauer, and Zwart-Gerber- Belamri cavitation models. Are these from the literature orverified by your experimental comparison?
Response 5: Thank you reviewers for your suggestions on cavitation model. The performance comparison of cavitation models mentioned in this paper is summarized on the basis of others' research.
Point 6:Figures 4, 6, and 10, require improvement so details are visible. Currently, it is not journal-quality.
Response 6: Thanks for the reviewer's suggestion, the figure has been reprocessed and replaced in the original text.
(a)
(b)
Figure 4.Experimental rig of the multiphase pump.(a)Schematic diagram(b) Impeller and diffuser
Figure 6. Cavitation characteristics.
SS PS
(a)
SS PS
(b)
SS PS
(c)
Figure 10. Velocity distribution at 0.5 span under IGVF=0,0.1,0.2: (a)IGVF=0; (b)IGVF=0.1; (c)IGVF=0.2.
Point 7:In the Introduction, the author discussed researchers exploring cavitation as it seriously affects the operation of the multiphase pump. Since cavitation plays detrimental effects commonly, | suggest also introducing the current experimental work on investigating the thermal effects of cavitation. Kindly add the reference in the Introduction, of [Energy (2022): 124426.; Ultrasonics Sonochemistry (2022): 106035.; Journal of Cleaner Production (2022): 130470.; International Journal of Heat and Mass Transfer 170 (2021): 120970.].
Response 7: Thanks for the suggestions of the reviewers. These references are very helpful to improve the quality of the article. I have added these articles to the introduction. And the relevant changes have been highlighted in red.
Point 8:Lines 304-306, how come“.. since the cavitation on the blade SS is larger than that on blade PS, the blade load corresponding to the cavitation region is reduced to 0..."?
Response 8: Thank you for your suggestion. I have removed this expression from the original text.
Point 9:In conclusion, there exists a strong need here to emphasize your unique findings. | suggest the author give a suggestion of the optimized operation condition of cavitation flow in the multiphase pump based on your results.
Response 9: Thanks for the reviewer's suggestion. According to the research findings, cavitation flow in the multiphase pump is rather complicated. In order to further avoid cavitation phenomenon, a convex structure can be set on the blade to improve the flow state in the pump.

Reviewer 2 Report
Comments and Suggestions for Authors
Please read the attachment. Thank you.

Author Response
Response to Reviewer 2 Comments
Point 1: Title: Please consider changing to“Analyzing the Inlet Gas Void Fraction on the Flow Characteristics for a Multiphase Pump Based on Cavitation Cases.
Response 1: Thanks for the reviewer's advice. I have modified the title according to the requirements.
Point 2:Keywords: Please provide between 5 and 10 keywords that should not repeat thewords/phrases that appeared on the manuscript title.
Response 2: Thank you for your suggestions. Cavitation evolution, cavitation stage, CFD, Velocity distribution and backflow vortex are added in Keywords.
Point 3:Error! Reference source not found: there are 14 errors in citations. Please find and correctt them. (Lines 183, 245, 271, 280, 290, 311-312, 325, 342, 353, 370-371, and 384).
Response 3: Thanks for your helpful comments from the reviewers. I have checked the full text in detail and revised the grammar and wording. The changes are shown in red in the text.
Point 4:Line 91: number 3 in the unit of the flow rate should be superscripted. Q=100 m3/h should be formatted as follow Q=100 m3.h-1.
Response 4: Thanks to the reviewers, I have revised the original text and marked it in red.
Point 5:All equations should be mentioned or explained in the text.
Response 5: Thanks for the reviewer's suggestion, relevant equations have been added. For example
The w equation is added. Other equations have been added and highlighted in red.
Point 6:There are two sections 3. Please revise. Lines 176-177: Please revise the last sentence, "The convergence accuracy is set to 10-5." 10-5 is the timestep or the accuracy. How could the authors set that value? The authors selected the turbulence model. Why did the authors not choose the SST K-epsilon turbulence model in this study? Please explain in the manuscript.
Response 6: Thanks to the reviewers, 10-5 is the convergence accuracy. When the convergence accuracy is 10-5, the numerical results are in good agreement with the experimental results. The k -epsilon turbulence model has weak results in predicting strongly separated flows, flows containing large curvature and flows with strong pressure gradient. Therefore, the SST k-w turbulence model was used for numerical simulation.
Point 7:CFD is not a novel method for field study. Why didn't the authors approach the optimization design for this kind of pump? Have you considered optimization design for your further studies? Please mention it as your additional task. You also introduce them in the literature review or the last paragraph of the conclusion section. The related works could be considered: CFD Analysis and Optimum Design for a Centrifugal Pump Using an Effectively Artificial Intelligent Algorithm; Centrifugal Pump Design: An Optimization.
Response 7: Thanks to the reviewers, we consider the optimization of the pump design. We use orthogonal optimization method to carry out the design. I have supplemented our optimization work. These two papers have also been cited and are very useful for improving the quality of the papers. Thanks again for the reviewers.
Point 8:The quality of some figures is poor; please increase their resolution. They are Figures 3-8 and 10.
Response 8: Thanks for the reviewer's suggestion. I have replaced the pictures with low resolution.
Point 9:References: Citation in the text and references should be followed the journal template.This research lacks connections, and the literature review of optimization design is limited. The reviewer suggests that you should search the Journal of Marine Science and Engineering or other Journals for more references that could be used to enrich your literature review. Generally, a manuscript (average) will have about 35-50 papers.
Response 9: Thanks for the reviewer's comments. I have supplemented the literature related to optimization and marked it in red.
Point 10:How did you set the timestep in this study? How many designed models did you use for the initial structure and size design?
Response 10: Thanks to the reviewers, the research in this paper is carried out under the steady-state condition, so the time step is not involved. In the initial design, many models were designed, about 40 or so, and the best were selected.
Point 11:Have the authors checked the MESH Quality? Please provide the ANSYS Mesh Metrics and Shape Checking chart in the manuscript.
Response 11: Thanks for the reviewer's comments. I have checked the grid quality, supplemented the data related to grid quality and marked it in red.
Point 12:How did the authors evaluate the validity of their results? And what are the main limitations of this approach?
Response 12: Thanks to the reviewers, we compared the experimental head and efficiency with the numerical results. One limitation of this method is that it does not compare the internal flow field.
